# Temporal Shift of the Respiratory Syncytial Virus Epidemic Peak for the Years 2020–2023 in Marseille, Southern France

**DOI:** 10.3390/v15081671

**Published:** 2023-07-31

**Authors:** Lucille Claire De Maria, Philippe Colson, Aurélie Morand, Noémie Vanel, Didier Stoupan, Bernard La Scola, Céline Boschi

**Affiliations:** 1IHU Méditerranée Infection, 19-21 Boulevard Jean Moulin, 13005 Marseille, France; lucille-claire.de-maria@ap-hm.fr (L.C.D.M.); didier.stoupan@ap-hm.fr (D.S.); bernard.la-scola@univ-amu.fr (B.L.S.); 2Assistance Publique-Hôpitaux de Marseille (AP-HM), 264 Rue Saint-Pierre, 13005 Marseille, France; 3Institut de Recherche Pour le Développement (IRD), Microbes Evolution Phylogeny and Infections (MEPHI), Campus Aix-Marseille Université, 27 Boulevard Jean Moulin, 13005 Marseille, France; 4Assistance Publique-Hôpitaux de Marseille (AP-HM), Hôpital Timone, Service des Urgences Pédiatriques, 264 Rue Saint Pierre, 13005 Marseille, France; aurelie.morand@ap-hm.fr; 5Assistance Publique-Hôpitaux de Marseille (AP-HM), Hôpital Timone, Service de Réanimation Pédiatrique de La Timone, 264 Rue Saint Pierre, 13005 Marseille, France; noemie.vanel@ap-hm.fr

**Keywords:** respiratory syncytial virus, temporal shift, France, Marseille, SARS-CoV-2

## Abstract

Respiratory syncytial virus is among the most common causes of respiratory infections. Typically, this viral infection has a seasonality during the cold months but with the SARS-CoV-2 pandemic this has been considerably modified. Here, we studied the epidemiology of this virus in university hospitals of Marseille, South of France, over the period 2020 to 2023. We tested in our laboratory from July 2020 to October 2021 16,516 nasopharyngeal swabs from 16,468 patients for RSV infection using different qPCR assays. We then analyzed data from previous and subsequent winters (from 2018 to 2023) and previous summers (from 2015 to 2021). A total of 676 patients were RSV-positive; their mean age was 3 years and 91 were under 5 years of age. We observed a delay of 4 months of the RSV epidemic’s onset compared to other years with an epidemic that peaked in March 2021. We had significantly more RSV-positive cases during summer 2021 compared to previous summers, whereas the incidence of RSV infections was not significantly higher during winter 2022 versus previous winters. Moreover, 494 patients were diagnosed as RSV-positive in the emergency unit and 181 were subsequently hospitalized, and 34 patients were diagnosed RSV-positive while already in the intensive care unit. Over all the study periods, 38 patients diagnosed as RSV-positive died, the majority of whom (23/28) were over 65 years of age. These data show an atypical evolution of the incidence of RSV infections in our city and is another example of the unpredictability of infectious disease epidemiology.

## 1. Introduction

Human orthopneumovirus, or respiratory syncytial virus (RSV), is one of the most common causes of respiratory infections in young children worldwide [1]. It was first described in 1956 as causing coryza in chimpanzees [2]. It was first isolated from humans in 1957 in Baltimore from the respiratory specimens of two children with broncho-pneumonia and laryngotracheobronchitis [3]. It was first called “chimpanzee coryza agent” (CAA), then renamed “respiratory syncytial virus” because of its cytopathic effect in tissue culture, which looks like a large cell-like structure known as a syncytium. It is a single-stranded, negative polarity, enveloped RNA virus that belongs to the family *Paramyxoviridae* and genus *Orthopneumovirus* with two major antigenic subgroups, A and B. These two genotypes can co-circulate during the same outbreak, but one often predominates. The World Health Organization (WHO) reports that this virus is globally responsible for more than 60% of acute respiratory infections among children. Based on the results of the Houston Family Study between 1975 and 1980 in the United States [4], it is now known that children will have at least one episode of RSV infection by the age of 2 year-old. RSV can affect both upper and lower respiratory tract illness with symptoms such as coughing, wheezing, or bronchiolitis with respiratory distress [5]. Children at increased risk of developing severe bronchiolitis are those under 6 weeks of age, those born prematurely, those with an underlying cardiopulmonary disorder, or those immunocompromised. This virus is also known to infect people over the age of 65 years and is responsible for pneumonia or exacerbation of chronic lung or heart disease [6]. Some studies suggested that during outbreaks between 2% and 20% of infections with RSV can lead to death among the elderly in nursing homes [7,8,9]. A rise in interest for RSV epidemiology is currently linked to the availability of monoclonal antibodies that are planned to be broadly used among children next winter and particularly nirsevimab at a broad scale [10,11].

Typically, under temperate climates as in France, RSV infections have a seasonality with an onset in late autumn and an ending in early spring, which corresponds on average to a 4-month epidemic. However, we observed during the winter of 2020/2021 in Marseille (Assistance Publique des Hôpitaux de Marseille (AP-HM), South of France) that the epidemiology of this virus had been strongly disturbed [12]. As a matter of fact, the seasonality and yearly incidence of viral respiratory infections appear much more complicated than can be reduced to an occurrence of cases during the cold months of the year. As a matter of fact, major epidemic or pandemic events such as the H1N1 pandemic in 2009 or the current SARS-CoV-2 pandemic have been associated with significant changes in the incidence of other viral respiratory infections. Regarding SARS-CoV-2, it emerged in December 2019 in China and in February 2020 in France and became pandemic in March 2020. The intensity of its spread and its clinical evolution, with 2,600,498 cases and 69,000 associated deaths for the year 2020 in France, has led to the implementation of several public health measures, including lockdown, curfew, social distancing, and mask wearing [12]. These measures and possible interactions between respiratory viral infections have been reported as associated with changes in the circulation of other respiratory viruses, but in a very variable manner depending on the virus. This has been widely reported for the case of influenza viruses worldwide and in our center [12]. Similarly, several teams have reported worldwide [13,14] and in France [15,16] a substantial change in the incidence of RSV infections during the winter of 2020–2021.

Here, we wanted to describe the incidence and main features of the RSV infections diagnosed in our center to figure out how far the RSV epidemiology was modified during the SARS-CoV-2 pandemic. To try to explain this epidemic time lag during this period, we retrospectively analyzed our data for summers since the year 2015. We also wanted to compare the data for the winters of 2021–2022 and 2022–2023 with those of previous winters to assess whether the incidence rebounded due to immunity loss.

## 2. Materials and Methods

We tested in our laboratory all nasopharyngeal swabs collected from patients in all public and university hospitals of Marseille (AP-HM), southeastern France from 1 July 2020 to 30 October 2022. Marseille is the second-largest city in France and accounts approximately for 870,000 inhabitants. AP-HM includes four university hospitals with approximately 3400 beds, 125,000 admissions, and one million consultations per year. Our laboratory is the single one for all diagnoses of infections for all AP-HM. We diagnosed RSV infections on nasopharyngeal swabs with real-time reverse transcription-PCR (qPCR). We used three different assays: (1) Fast Track Diagnosis Respiratory pathogens 21 (Fast Track Diagnosis, Luxembourg); (2) FilmArray Respiratory panel 2 plus (Biomérieux, Marcy-l’Etoile, France); and (3) GeneXpert Xpert Flu/RSV (Cepheid, Sunnyvale, CA, USA), according to the manufacturers’ instructions. These three techniques were used concomitantly throughout the year according to clinicians’ prescriptions. The first test is used at the core laboratory and is performed once a day, with a time to result delivery of approximately 24 h. The second and third tests were performed at the point-of-care laboratories following clinicians’ requests for an urgent result report and their will to test for 21 or 4 pathogens, respectively.

We kept in our study only the nasopharyngeal swabs collected from the same patients at least 7 days apart when there were several samples. We retrospectively analyzed the number of samples and of those RSV-positive for all months between May and August (hot months) for each of the years from 2015 until 2021, as well as for all months from October until January for each cold season between the end of 2018 and the beginning of 2022, to determine if differences occurred between years before and during the SARS-CoV-2 pandemic. Finally, we studied over all the periods of the present study (2015–2023) how many RSV-positive patients were hospitalized after being admitted in the emergency ward, in which type of clinical wards they were hospitalized, and how many RSV-positive patients died.

All data were generated as part of the routine work at AP-HM, and this study results from routine standard clinical management. Access to the patients’ biological and registry data issued from the hospital information system and to pathogens detected in deceased patients was approved by the data protection committee of AP-HM and was recorded in the European General Data Protection Regulation registry under number RGPD/APHM 2019-73. We were also able to retrieve data retrospectively thanks to our computerized tool Epimic for epidemiological surveillance and alerts based on microbiological data, developed in our laboratory in 2002 [17].

Figures were plotted with Graphpad.9 and statistical analyses were carried out using the OpenEpi version 3.01 online tool accessed on 6 April 2001 (https://www.openepi.com/) using *p* = 0.05 as the significance threshold.

## 3. Results

From 1 July 2020 to 30 October 2021, we screened for RSV 16,516 nasopharyngeal swabs collected from 16,468 patients and we obtained a positive diagnosis for 702 nasopharyngeal swabs (4.2%) from 676 patients (4.1%). These patients were 357 men and 319 women (sex ratio M/F, 1.2). Their mean age was 3 years (range from 16 days to 96 years). It is worth noting that 90.5% (*n* = 612) of the RSV-positive patients were under 5 years of age. Forty-one RSV-positive patients were between 5 and 15 years old (6.1%), 19 patients were between 15 and 65 years old (2.8%), and only 4 patients were over 65 years old (0.6%).

First, we analyzed the evolution of the incidence of RSV diagnoses from July 2020 to October 2021. We observed only two RSV-positive cases in December 2020 and then 339 cases between January and April 2021, when the epidemic reached its peak. We then diagnosed 97 cases during spring and summer (May to August) 2021. We had only seven RSV-positive patients between 1 September and 15 September. Then, from 15 September 2021 to 30 October 2021 we observed an increase in the number of cases with 231 RSV-positive patients, corresponding to the beginning of the RSV epidemic for winter 2021–2022. Hence, two consecutive epidemics occurred during the year 2021, separated by a period of low RSV incidence (Figure 1).

Second, we retrospectively analyzed the numbers of clinical samples tested and of those diagnosed as RSV-positive during the previous summers (May to August) to figure out if we had a significant increase in positive cases for our study period (Figure 2). We indeed observed a significant increase in incidence for this May–August period for the year 2021 (2.6%, 97 patients positive/3675 patients tested) vs. the years 2015 to 2020, for which we had respectively 0% (0/1097), 0.7% (8/1154), 0.5% (6/1218), 0% (0/1780), 0.4% (9/2176), and 0% (0/3384) RSV-positive patients (*p* < 0.001 vs. all the other years) (Figure 2).

Third, we studied if the number of RSV diagnoses increased following the period of very low incidence as a possible consequence of immunity drop. Although no case was diagnosed during the cold months (October–January period) of 2020–2021 (0 RSV-positive patients out of 5597 patients tested), the proportion of RSV-positive patients among those tested was not significantly higher during the subsequent same period of 2021–2022 (752 RSV-positive patients out of 6472 patients tested; 11.6%) and of 2022–2023 (1864/13,980; 13.3%) compared to the same previous periods of 2018–2019 (925/5857; 15.8%) and 2019–2020 (929/6795; 13.6%) (Table 1). As a matter of fact, the numbers of tested patients and RSV-positive patients in October 2021–January 2022 were 1.02 and 0.81-fold those in October 2018–January 2019 and in October 2019–January 2020 combined, respectively. In October 2022–January 2023, the numbers of tested patients and RSV-positive patients were 2.16 and 2.48-fold those in October 2018–January 2019 and in October 2019–January 2020 combined, indicating that the higher number of positives was likely the consequence of an increasing number of tests.

Fourth, we looked at the available data about hospitalizations, admission in intensive care units, and RSV diagnosed in patients who died. From July 2020 to October, 494 patients (73%) were diagnosed as RSV-positive during their stay in the emergency ward. Of them, 181 (37%) were subsequently hospitalized following this diagnosis. A majority were children between 16 days and 12 years old (mean age: 0.7 years), while two were adults aged 36 and 64 years. Moreover, 34 patients were diagnosed with RSV infection while already in the intensive care unit. Regarding the mortality of RSV-positive patients, regardless of the imputability or not of RSV infection in a patient’s death, which was not studied here, we observed no RSV-associated deaths for the October–January and May–August periods of 2020–2021. Over the same subsequent winter period, two patients died in 2021–2022 and ten patients died in 2022–2023. If we compare this to the winters of 2018–2019 and 2019–2020, there is no significant difference in terms of mortality, as we had eight deaths in each of the two periods. We analyzed the age of patients who died over the different periods and the majority (23 out of 28 patients) were over 65 years of age. For the 2018–2019 period, the average age was 84 years (min: 73 years; max: 95 years). For the 2019–2020 period, among the eight patients who died, there was a 1-year-old child and a 37-year-old adult, and the other six patients had an average age of 82 years (min: 70; max: 93). For the period 2021–2022, two 52- and 58-year-old patients died. Finally, of the ten patients who died in 2022–2023, there was one child aged 6 years, one adult aged 54 years, and the other eight patients had an average age of 82 years (min: 66 years; max: 98 years). Twelve of these patients were in the intensive care unit, seven were hospitalized in other clinical wards, and eight went initially to the emergency room.

## 4. Discussion

First, we observed dramatic changes in the seasonality of RSV infection. We were here able to describe RSV incidence over 15 months during the SARS-CoV-2 pandemic in university and public hospitals. In addition, we could compare the numbers of diagnoses and deaths of RSV infections during the hot and cold months over more than five years. We had only diagnosed a few cases of RSV infection in our university hospital laboratory during the 2020 winter in Marseille, despite a high number of tests performed in our laboratory, while strict lockdown measures for SARS-CoV-2 were in place in France [12]. This phenomenon has been widely described in France and in other countries and we then observed a delay of 4 months for the onset of the RSV epidemic compared to the previous years. This unusual delay was also observed in other regions of France [15,16] as well as in other countries such as Australia, South Africa, the United States, Brazil, Chile, Japan, Canada, and South Korea [13,14]. The epidemic in our city extended from March to July 2021 with cases still occurring later during summer 2021. When we investigated in the present study the number of RSV diagnoses during summer 2021, we observed a significant increase in the percentage of positive cases (2.6%) compared to previous years, indicating that the increase in the number of cases was not related to a greater number of tests performed. Summer epidemics have already been described before the COVID-19 pandemic, notably in the USA in Minnesota during the summer of 2017 [18]. The fact that this delay of the RSV epidemic onset has been observed on a global scale raises many questions about its precise cause. Several hypotheses have currently been put forward that primarily involve the SARS-CoV-2 pandemic. They notably consist, first, of the establishment of public health measures introduced in our country in March 2020 that would have prevented the virus from circulating, particularly in young children [15]. We have previously described a fall in the incidence of respiratory viruses that cause lower-tract respiratory infections, while the incidence of other respiratory viruses was not significantly decreased [12]. A second hypothesis is viral interference: the SARS-CoV-2 pandemic could have prevented the RSV epidemic due to a competition between the two viruses [19,20]. Multiplex PCR diagnostic analysis, which has been widely developed in recent years, allows the rapid and simultaneous diagnosis of several viruses at the same time. It is therefore interesting to look at the interaction between respiratory viruses to better understand if there are exclusion or viral interference mechanisms that could explain why some viruses circulate with alternative incidence, as was the case during our study between RVS and SARS-CoV-2, for example. It is also notable that a similar situation had already been observed in 2009, when the delay of the RSV epidemic was explained by the influenza A/H1N1 2009 pandemic that raged during that year [21].

Second, our data do not indicate that RSV immunity was lost due to a dramatically lower incidence and a temporal shift of the yearly epidemic period. Indeed, our epidemiological data revealed that there were few cases of RSV infections during the cold months of 2019–2020, which might have led to a loss of immunity toward this virus, as reported by den Hartog et al. [22]. This loss of immunity, which can be observed through antibody testing, raises the question: what about future RSV outbreaks [23]? In this view, it is noteworthy that the analysis of our data from the cold months of 2021–2022 and 2022–2023 (October to January) shows that in our geographical area and university hospital setting, there were not more RSV-diagnosed cases than in the winters before the COVID-19 pandemic. As a matter of fact, the number of positive patients rose 2.5-fold during the cold months of 2022–2023, but this was in the setting of a 2.2-fold rise in the number of tests. During the SARS-CoV-2 pandemic, there was a broader use of multiplex PCR, which allows several respiratory viruses to be tested at the same time. There was indeed a growing number of requests by clinicians for multiplex PCR testing, considering that some allow a rapid report of results and the knowledge that it is sometimes difficult to distinguish based on clinical symptoms only between respiratory viruses and that several respiratory viruses co-circulate during the whole year and can cause co-infections. Also, at the laboratory level, the use of multiplex PCR testing eased the management of the large number of respiratory samples and the completion of the virological diagnoses.

Third, we were able to determine that RSV-related deaths were essentially in elderly people during all study periods. Indeed, 23 out 28 patients who died were over 65 years old. It is currently admitted that RSV infections are severe not only in children but also in elderly patients (especially people over 65 years old, those with cardiopulmonary problems, and those with immunodeficiency [6]). A recent meta-analysis concluded that the disease burden associated with RSV in elderly patients was higher than previously described in the literature [24].

Our study has some limitations. One is its university hospital setting, which does not allow an appreciation of the RSV epidemic in the whole population of our geographical area. Another one is the absence of study of the imputability of RSV infections in the death of RSV-positive patients, as we only report here on the association between RSV diagnosis and death, without speculating on the causality link. An imputability study would require access to the patients’ clinical records and a specific study. Also, we did not determine if RSV was of genotype RSV-A or RSV-B, either with qPCR or with next-generation sequencing. Through sequencing, the state of Minnesota in the Unites States was able to describe the emergence of a new RSV lineage during an outbreak in the summer of 2017 [18]. Otherwise, for the winter of 2022–2023 we did not diagnose a greater proportion of RSV infections than during previous winters in a setting of approximately twice as many tests, considering that our study stopped in January 2023. However, some studies have already reported an increase in RSV cases compared to previous winters. A recent study in Austria [25] found that the number of RSV circulating in winter 2022 was higher than during previous years. They observed a change in the circulation and thus in the epidemiology of circulating RSV strains, as the RSV-A GA 2.3.5 lineage was predominant in 2021 and the RSV-B GB5.0.5a lineage was predominant in winter 2022. More interestingly, they found that this lineage was already circulating before the COVID-19 pandemic and that it was not a new, more virulent emerging lineage. A study in the state of Massachusetts in the Boston [26] area also reported a surge in RSV cases and genomic studies showed that this increase in RSV was due to the co-circulation of RSV-A and RSV-B, with a clear predominance of RSV-A (91%) over RSV-B (9%). Finally, we have not analyzed the potentially associated viral co-infections for RSV-positive patients.

## 5. Conclusions

The present data combined with previous findings show an atypical evolution of the incidence of RSV infections in the course of the SARS-Co-2 pandemic. They are a new example of the unpredictability of respiratory viral infections. The public health measures to reduce SARS-CoV-2 transmission probably had at least a partial impact on the delayed onset of the RSV epidemic to spring 2021. Still, the causes are probably multifactorial. In addition, the impact of epidemiological changes on people’s immunity to RSV at the population scale is unclear. The broad use of monoclonal antibodies in the very near future will beyond any doubt provide new additional keys on the understanding of the immune responses to RSV. Finally, very few studies to date have investigated the change in the genotype (A or B) of RSV during the period of the COVID-19 pandemic [25]. The genotype of this virus could allow a better understanding of the global evolution of its incidence and its assessment deserves future molecular epidemiology studies.

## Figures and Tables

**Figure 1 viruses-15-01671-f001:**
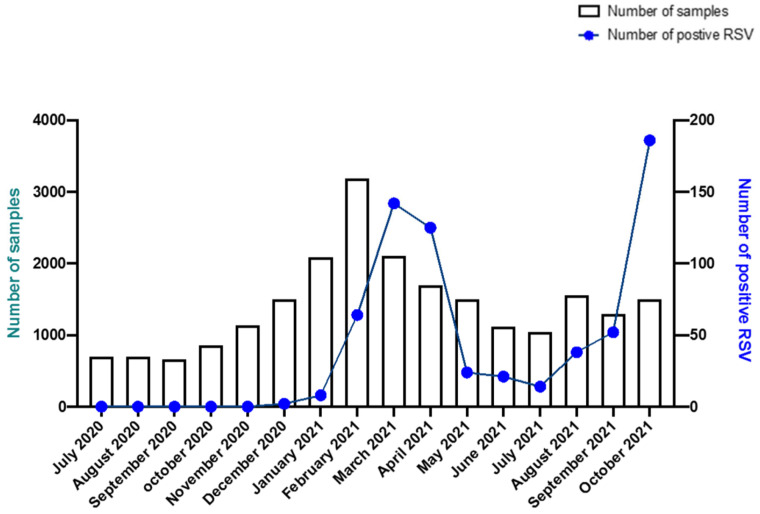
Distribution of number of samples and positive patients for RSV between July 2020 and October 2021. Left *Y*-axis represents the number of tested patients (bars); right *Y*-axis represents the number of RSV-positive patients (blue curve); *X*-axis represents time (months) from July 2020 to October 2021.

**Figure 2 viruses-15-01671-f002:**
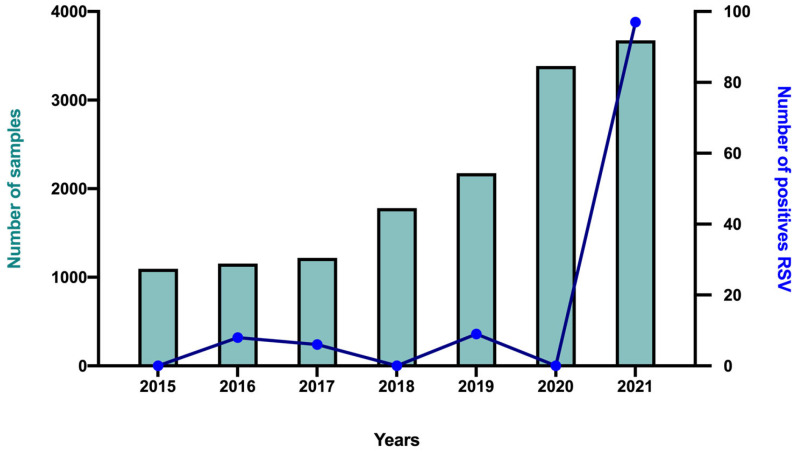
Distribution of number of samples and positive patients for RSV between May and August for each year. Left *Y*-axis represents the number of tested patients (green bars); right *Y*-axis represents the number of RSV-positive patients (blue curve); *X*-axis represents time (months) between May and August for each year between 2015 and 2021.

**Table 1 viruses-15-01671-t001:** Number of tested patients for RSV and RSV-positive patients between October and January for each year between 2018 and 2023.

Period	Number of Tested Patients (N)	RSV-Positive Patients (N, %)
October 2018–January 2019	5857	925 (15.8)
October 2019–January 2020	6795	929 (13.7)
October 2020–January 2021	5597	0 (0.0)
October 2021–January 2022	6472	752 (11.6)
October 2022–January 2023	13,980	1864 (13.3)

## Data Availability

Not applicable.

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
