# Peer review of "Temporal Shift of the Respiratory Syncytial Virus Epidemic Peak for the Years 2020–2023 in Marseille, Southern France"

_viruses, 2023, doi:10.3390/v15081671_

Round 1

Reviewer 1 Report

De Maria et al. wrote a very interesting paper about RSV modified peak during the pandemic period.

RSV seasonality is well described, but pandemic, and its French consequences, lockdown, modified this virologic landscape. It is very interesting to analyze these changes, before the immunoprophylaxis by Nirsevimab would arrive in the next winter seasons.

16516 nasopharyngeal swabs were analyzed by qPCR assays between July 2020 and October 2021, and compared to seasons from previous years. A delay of 4 months was observed for the epidemic onset compared to other winters.

I would have few questions or remarks:

-       I think that the result of 90.5% of RSV-positive samples detected in children less than 5 years of age is important in the abstract.

-       In the abstract: “38 people diagnosed RSV-positive died”: it is important to detail the age (mean 76 years old).

-       In material and methods (lines 88-90): why are there 3 different methods of qPCR? Which one detected which period?

-       Table 1 and comments: maybe it would be clearer to analyze why the number of samples is doubled in the period October 2022-january 2023

-       Maybe it would be interesting to know if the age of the patients were different between the epidemic seasons.

Reviewer 2 Report

Respiratory syncytial virus (RSV) infection is a leading cause of severe lower track respiratory diseases in young children, older adults, and immunocompromised individuals worldwide. This paper reported the data of nasopharyngeal swabs for all public university hospitals of Marseille (AP-HM), southeastern France from 01/07/2020 to 30/10/2022. The authors also retrospectively analyzed the number of samples and of those RSV-positive for all months between May and August (hot months), for each of the years from 2015 until 2021, as well as for all months from October until January for each cold seasons between end of 2018 and start of 2022, to determine if differences occurred between years before and during the SARS-CoV-2 pandemic. From the data, they observed a delay of 4 months of the RSV epidemic onset compared to other years with an epidemic that reached its peak in March 2021 and significantly more RSV-positive cases during summer 2021 compared to previous summers. However, the conclusion and the significance of this paper are not clear.

1.        The introduction of this paper requires improvement to provide a comprehensive background on the topic. It should include a discussion on prior research, the thesis statement, and highlight the significance of the paper.

2.        The methodology section needs to incorporate additional details about the methods employed in this study. The authors mentioned the utilization of three different assays, but it is essential to explain why these specific assays were chosen and whether they produced consistent results. Furthermore, the authors compared data from this paper with previous and subsequent periods (2015-2023), and it is necessary to clarify the source of this data and the methodology employed for the comparison.

3.        The results section presents information on gender, age, and death rate; however, it lacks clarity, making it difficult to discern the main points. Additionally, Figures 1 and 2 are insufficiently illustrated and fail to provide a clear conclusion. In Figure 2, are the RSV positive numbers for 2015, 2018, and 2020 zero?

4.        The conclusions section fails to articulate the precise findings of this paper, leaving the reader unsure about the key takeaways. Moreover, the significance of this research is not adequately clarified, and it would benefit from a more explicit explanation.

The English language should be improved for a better presentation.

Reviewer 3 Report

The paper reports changes in the circulation of  RSV  in during  Covid pandemia  in sud of France   during the winter of 2020-2021. All nasopharyngeal swabs for all public university hospitals of Marseille, southeastern France from 01/07/2020 to 30/10/2022 were analyzed  and diagnosed RSV infections on nasopharyngeal swabs by real-time reverse transcription-PCR (qPCR).,Results   were analized and  authors  reports that , two consecutive epidemics occurred during year 2021, separated by a period of low RSV incidence 

Except for a one-year-old child and a 38-year-old adult,  all RSV-positive patients who died were elderly people.  the burden disease associated with RSV in  elderly patients was higher than previously described in the literature . Similar data are previusly reported from others study( ie Torres AR, Guiomar RG, Verdasca N, Melo A, Rodrigues AP; Rede Portuguesa de Laboratórios para o Diagnóstico da Gripe. Resurgence of Respiratory Syncytial Virus in Children: An Out-of-Season Epidemic in Portugal. Acta Med Port. 2023   Jan 27. and Savic M, Penders Y, Shi T, Branche A, Pirçon JY. Respiratory syncytial virus disease burden in adults aged 60 years and older in high-income countries: A systematic literature review and meta-analysis. Influenza Other Respir Viruses. 2023 Jan;17(1):e13031)

. . But the paper is  well organized and the population in study is important . 

 I think it may be accepted  with low priority 
